# An Efficient Cyan Emission from Copper (II) Complexes with Mixed Organic Conjugate Ligands

**DOI:** 10.3390/ma15051719

**Published:** 2022-02-25

**Authors:** Jingjing Wang, Junjie Ren, Qinglin Tang, Xinzhi Wang, Yao Wang, Yanxin Wang, Zhonglin Du, Wei Wang, Linjun Huang, Laurence A. Belfiore, Jianguo Tang

**Affiliations:** 1Institute of Hybrid Materials, National Center of International Joint Research for Hybrid Materials Technology, National Base of International Sci. & Tech. Cooperation on Hybrid Materials, Qingdao University, 308 Ningxia Road, Qingdao 266071, China; w1834089@163.com (J.W.); m2019025747@163.com (J.R.); a15666920912@163.com (Q.T.); wangxinzhi1988123@163.com (X.W.); wangyaoqdu@126.com (Y.W.); yanxin_2008@126.com (Y.W.); duzhonglin@qdu.edu.cn (Z.D.); wangwei040901@163.com (W.W.); huanglinjun@qdu.edu.cn (L.H.); belfiore@engr.colostate.edu (L.A.B.); 2Department of Chemical and Biological Engineering, Colorado State University, Fort Collins, CO 80523, USA

**Keywords:** copper (II) complex, luminescent materials, benzylimidazole, bipyridine, salicylic acid, spectroscopy

## Abstract

Copper (II) complexes containing mixed ligands were synthesized in dimethyl formamide (DMF). The intense cyan emission at an ambient temperature is observed for solid copper (II) complexes with salicylic acid and a 12% quantum yield with a fluorescent lifetime of approximately 10 ms. Hence, copper (II) complexes with salicylic acid are excellent candidates for photoactive materials. Fourier transform infrared spectroscopy (FTIR) and X-ray photoelectron spectroscopy (XPS) reveal that the divalent copper metal centers coordinate with the nitrogen and oxygen lone pairs of conjugate ligands. XPS binding energy trends for core electrons in lower-lying orbitals are similar for all three copper (II) complexes: nitrogen 1s and oxygen 1s binding energies increase relative to those for undiluted ligands, and copper 2p_3/2_ binding energies decrease relative to that for CuCl_2_. The thermal behavior of these copper complexes reveals that the thermal stability is characterized by the following pattern: Cu(1,10-phenanthroline)(salicylic acid) > Cu(1,10-phenanthroline)(2,2’-bipyridine) > Cu(1,10-phenanthroline)(1-benzylimidazole)_2_.

## 1. Introduction

Transition metal ions and complexes have attracted much attention due to their incompletely occupied d-orbitals and highly ordered solid-state structures [1]. These complexes possess unique optical and magnetic properties arising from unpaired d-electrons and a non-zero spin angular momentum. Although some examples of complexes with high-spin ground states and ordered structures have been achieved (Mn) [2,3,4,5], the rational design and synthesis of novel transition metal complexes remains a challenge for luminescent sensors, photo-luminescent devices, and biological probes [6,7,8,9].

To date, a variety of noble metal complexes have been developed (mainly Au, Ag, and Pt) [10,11,12,13] due to their chemical stability and facile synthetic procedures. However, the commercial development of noble metal ions requires further consideration because of their toxic environmental impact. The applications of transition metal complexes exhibit limitations due to the cost of these precious metals and their toxicity. Compared to noble metal s(Au, Ir, and Ru), divalent copper is relatively abundant [14,15], significantly cheaper, has a high thermal stability [16,17,18], and is readily available from commercial sources. Hence, Cu^2+^ complexes appear to be excellent candidates for this research investigation.

The electronic configuration of divalent copper ions is [Ar]3d^9^ or [Ar]4s^1^3d^8^, with incompletely filled d-orbitals and relatively low-energy d-d transitions. Luminescent emission occurs if radiative pathways exist from the exited state to the ground state with a significant quantum yield. However, the free-ion excited states of Cu^2+^ experience ultra-fast non-radiative processes, such that Cu^2+^ complexes are not advantageous from a photo-physical point of view [19]. Due to the influence of organic ligands that quench emission, the luminescent studies of Cu^2+^ complexes are very scarce in the research literature [20,21,22]. Hence, this investigation of Cu^2+^ complexes, with a high quantum yield, is very challenging. Recently, successful research on copper (II) luminescence has revealed that phenanthroline (i.e., phen) is a promising ligand that does not quench emission [23,24,25]. The most common preparation method for Cu^2+^ complexes is via the self-assembly of the metal ion center with the appropriate organic ligands to promote the formation of extended framework structures [25,26]. Hence, current research focuses on the synthesis of copper (II) complexes with extended organic frameworks. Organic ligands that contain oxygen or nitrogen donors play important roles in the construction of coordination complexes [27,28]. Previous luminescent results for copper (II) complexes, and their main photo-physical parameters, are listed in Table 1 [25,29,30]. These results provide guidelines for future research on copper (II) complexes that exhibit luminescent emissions with a significant quantum yield and measurable lifetimes. This research investigation extends the results in Table 1 to copper (II) complexes that contain bidentate organic conjugate ligands.

Unlike luminescent emission in trivalent lanthanide complexes, Cu^2+^ emission wavelengths are influenced strongly by the electron donating/withdrawing ability of ligands in the first-shell coordination sphere, as well as organic conjugate ligands, solvents, crystal structures, and pH levels. As indicated in Table 1, visible emission occurs in the range from 450 nm (i.e., blue) to 522 nm (i.e., green). The ligand choice and structure provides some control with respect to copper (II) emission wavelengths in solid-state fluorescent studies.

Combining structural and luminescent factors, the primary objective of this research investigation is to obtain an enhanced blue-green emission of copper (II) complexes that contain favorable conjugate ligands with a light-harvesting ligand-to-metal energy transfer. Previous research in our laboratory focused on several metal–ligand frameworks based on nitrogen- and oxygen-containing organic ligands for lanthanide emission [8,32]. In this investigation of copper (II) emission, the following three Cu^2+^ complexes were designed and synthesized in DMF: A = Cu(phen)(bim)_2_, B = Cu(phen)(bipy), and C = Cu(phen)(SA), where bim = 1-benzylimidazole, bipy = 2,2’-bipyridine, SA = salicylic acid, and phen = 1,10-phenanthroline. Results presented herein demonstrate that the appropriate design strategies can deliver new phenanthroline-based materials for their use as luminescent materials. In the DMF solution, these complexes emit a cyan color with a significant quantum yield. From an application viewpoint, the cyan emission spectra overlap with carotene absorption [33]. β-carotene is a key compound in grain plants, fruit trees, and other key food resource plants, which absorb solar energy at wavelengths between 500 nm and 550 nm. Hence, the significance for agricultural harvest improvement is very important.

## 2. Materials and Methods

### 2.1. Chemicals 

All analytical reagents were purchased from commercial sources and used without further purification. The details are as follows: 1-benzylimidazole (bim), 2,2’-bypyridyl (bipy), salicylic acid (SA), the antenna co-ligand 1,10-phenanthroline (phen), and analytical grade solvent N,N-dimethylformide (DMF) were purchased from Aladdin (Shanghai, China). Copper chloride dehydrate (CuCl_2_·2H_2_O) (98%) was purchased from R&M Chemical (Shanghai, China).

### 2.2. Material Preparation

**Synthesis of Complex A: Cu(bim)_2_(phen).** Complex A was prepared via a mixture of CuCl_2_·2H_2_O (17 mg, 0.10 mmol), bim (31 mg, 0.20 mmol), whose structure of the bim is as shown in Figure 1a, and phen (19 mg, 0.10 mmol) in 5 mL of DMF, which was stirred for 6 h at 30 °C. A dark blue turbid liquid was then obtained after 6 h and it was centrifuged to obtain the desired material. The structure of Complex A is shown in Figure 1d. The elemental analysis (%) and calc (exp) for Complex A is: C 65.56 (67.72), Cl 11.83 (6.26), N 12.75 (14.81), Cu 9.86 (11.2), and yield (63%, 32.76 mg). The IR (cm^−1^): (C=N: 1429 cm^−1^, Cu-N: 510 cm^−1^). From Figure 2a, the 1H NMR spectrum of Complex A in DMSO-*d*6 solution confirms its structure. The imidazole protons generally give a signal between 7.8 and 8.1 ppm in DMSO solution.

**Synthesis of Complex B: Cu(bipy)(phen).** Complex B was synthesized via the same procedure as Complex A, except that 1-benzylimidazole was replaced by 2,2’-bypyridyl with the structure shown in Figure 1b. A dark blue turbid liquid was then obtained after 6 h and it was centrifuged to obtain the desired material. The structure of the Complex B is shown in Figure 1e. The elemental analysis (%) and calc (exp) for Complex B is: C 52.71 (58.09), Cl 19.51 (15.63), N 12.06 (12.32), Cu 15.72 (13.97), and yield (38%, 25.46 mg). The IR (cm^−1^): (C=N: 1416 cm^−1^, Cu-N: 435 cm^−1^). Figure 2 reveals that the 1H NMR spectrum of Complex B exhibits a signal at between 7.9 and 8.1 ppm in DMSO solution.

**Synthesis of Complex C: Cu(SA)(phen).** Complex C was also synthesized via the same procedure as Complex A, except that 1-benzylimidazole was replaced by salicylic acid with the structure shown in Figure 1c. A dark blue turbid liquid was then obtained after 6 h and it was centrifuged to obtain the desired material. The structure of the Complex C is shown in Figure 1f. The elemental analysis (%) and calc (exp) for Complex C is: C 50.07 (52), Cl 21.76 (16.19), N 10.84 (6.38), Cu 16.73 (14.48), O 0.6 (10.95), and yield (76%, 38 mg). The IR (cm^−1^): (C=N: 1416 cm^−1^, Cu-N: 487 cm^−1^ Cu-O: 432 cm^−1^). Figure 2 reveals that the hydroxyl proton NMR signal of salicylic acid is observed at a larger chemical shift (i.e., 6.8–7.5 ppm) because hydrogen bonding affects this resonance in Complex C.

### 2.3. Characterization

Infrared spectra were recorded on a Perkin Elmer 400 Fourier transform spectrometer (FTIR) between 4000–400 cm^−1^. The ^1^H NMR spectra were measured in DMSO-*d*6 solutions on the Bruker Av HD 400 spectrophotometer (400 MHz), using TMS as an internal reference. XPS spectra were obtained by a XSAM-800 Kratos spectrometer. Al (Kα at 1486.6 eV) or Mg (Kα) radiation was used to excite photoelectrons from Cu (2p_3/2_), O (1s), and N (1s) core levels. The UV–Vis spectra were measured on a UV755B (Shanghai Youke Instrument Co., Ltd., Shanghai, China) spectrophotometer at an ambient temperature. The redox potentials E_pCuII/CuI_ were measured in the presence of the CHI-832 electrochemical workstation in 0.1 M KCl by using it as a supporting electrolyte for these complexes (0.02 M aqueous solutions). The voltage was scanned from −0.65 to 0.5 V with a scan rate of 100 mVs^−1^, with glassy carbon, Pt foil, and Ag/AgCl as working, auxiliary, and reference electrodes, respectively. Fluorescent absorption and emission spectra were measured on a Cary Eclipse fluorescence spectrophotometer (Varian, San Francisco, CA, USA) at an ambient temperature. A thermo-gravimetric analysis (SII TG/DTA 6300) was carried out using a Perkin Elmer TGA 4000 thermogravimetric analyzer. Emission scanning electron microscopy (SEM) images and energy dispersive X-ray (EDX) mapping were recorded on a JSM-6700F field emission scanning microscope (FEI, Czech, The Netherlands).

Figure 3 is a schematic diagram of the coordination of three different copper (II) complexes. All three copper complexes have two kinds of ligands. The first is the co-ligand: phen, and then the species of the second ligand is changed, which is (a) benzylimidazole, (b) bipyridine, and (c) salicylic acid, thereby forming three new copper (II) complexes.

## 3. Results and Discussion

### 3.1. Structures of the Coordination Complexes

The metal–ligand binding in copper (II) complexes was probed via infrared spectra for free ligands and complexes from 400 to 4000 cm^−1^. The data in Figure 4 reveal that the infrared spectra of three copper (II) complexes are similar to those of the corresponding ligands. The main infrared features in Figure 4 are due to absorptions of benzylimidazole (spectra 4a), bipyridine (spectra 4b), and the carbonyl group of salicylic acid (spectra 4c). There are some significant shifts in absorption frequencies of important ligand functional groups when these ligands coordinate with copper (II). For example, the imidazole C=N stretch absorbs at 1429 cm^−1^ (upper spectrum in 4a), representing a 12 cm^−1^ shift to a lower frequency in the copper (II) complex with benzylimidazole. This is attributed to copper (II) coordination with the imidazole nitrogen lone pair. In spectra 4b, the carbon–nitrogen in-plane stretch of bipyridine at 1442 cm^−1^ shifts to 1416 cm^−1^, possibly due to bidentate chelation with copper (II), as illustrated in Figure 4 for Complex B. Copper (II) complexes, with salicylic acid in spectra 4c, reveal a weak shift of the carbonyl stretch from 1658 cm^−1^ to 1652 cm^−1^, suggesting that the bonding strength of these copper (II) complexes varies as follows: bipyridine > benzylimidazole > salicylic acid. New infrared absorptions for all three copper (II) complexes near 500 cm^−1^ are not observed when copper (II) is absent. For example, copper (II)–nitrogen bonding is probably responsible for the 510 cm^−1^ absorption in complexes with benzylimidazole (spectra 3a), 435 cm^−1^ absorption in complexes with bipyridine (spectra 2b), and the 432 cm^−1^ absorption in complexes with salicylic acid (spectra 4c). The copper (II)–oxygen stretch at 487 cm^−1^ in the upper spectrum of Figure 3c suggests that the carboxylic acid group of salicylic acid coordinates with copper (II) in a bidentate fashion [26,31], as illustrated in Figure 4 for Complex C.

X-ray diffractograms of the three copper (II) complexes in Figure 4d suggest the formation of a well-defined crystal structure in each case. Diffractograms in Figure 4d were measured from 2θ = 5° to 60°. The strongest reflection for copper (II) complexes with benzylimidazole (i.e., upper X-ray diffractogram of Figure 4d) occurs at 2θ ≈ 10^0^, corresponding to an inter-planar lattice spacing of ≈8.8 Å. Two strong Bragg reflections are observed for copper (II) complexes with bipyridine (i.e., the middle X-ray diffractogram of Figure 4d) and salicylic acid (i.e., the lower X-ray diffractogram of Figure 4d) at essentially the same scattering angles in each case: 2θ ≈ 11^0^ and 28^0^, corresponding to the inter-planar lattice spacings of 8.0 Å and 3.2 Å, respectively.

X-ray photoelectron spectroscopy (XPS) yields information about metal–ligand interactions in these copper (II) complexes. Core electron binding energies for nitrogen 1s, oxygen 1s, and copper 2p_3/2_ are illustrated in Figure 5 for all three copper (II) complexes.

Nitrogen 1s binding energies in copper (II) complexes with benzylimidazole are 0.25 eV higher (i.e., 399.57 eV vs. 399.32 eV) relative to those in undiluted benzylimidazole, as illustrated in Figure 5b. A slightly smaller but similar shift in nitrogen 1s binding energies is observed in copper (II) complexes with bipyridine (i.e., 399.57 eV vs. 399.42 eV), according to Figure 5e. In copper (II) complexes with salicylic acid, oxygen 1s binding energies are 1.50 eV higher (i.e., 534.62 eV vs. 533.12 eV), relative to those in undiluted salicylic acid, as illustrated in Figure 5h. Two different oxygen 1s binding energies are resolved in Figure 5h for undiluted salicylic acid, but they are not resolved in the copper (II) complex. For all three copper (II) complexes investigated herein, the higher intensity peak for copper 2p_3/2_ electrons near 935 eV, due to spin-orbit coupling, reveals that binding energies are 0.20 eV to 0.55 eV smaller than the copper 2p_3/2_ binding energy of CuCl_2_ (i.e., 934.67 eV) according to Figure 5a,d,g. The copper 2p_3/2_ binding energies are the largest for bipyridine complexes (i.e., 934.47 eV) in Figure 5d and the smallest for salicylic acid complexes (i.e., 934.12 eV) in Figure 5g. XPS binding energy trends for core electrons in lower-lying orbitals are similar for all three copper (II) complexes: nitrogen 1s and oxygen 1s binding energies increase relative to those for undiluted ligands, and copper 2p_3/2_ binding energies decrease relative to that for CuCl_2_ [34,35,36].

A SEM analysis reveals that the ligand coordination with Cu^2+^ considerably modifies the surface morphology of the three complexes. STEM and EDX mapping images of prepared Cu^2+^ complexes are presented in Figure 6, demonstrating that N (green), O (yellow), and Cu (purple) elements are homogeneously distributed throughout the surface layer. This observation suggests that, in the vicinity of the surface layer, Cu^2+^ coordinates with all of the ligands employed in this study [37]. From Table 2, the content of each element in the complex is seen.

The content of each element in the complexes can be seen from Table 2. The element of the complexes contains copper, nitrogen, and oxygen.

### 3.2. Fluorescent Behavior of the Complexes

Figure 7 illustrates the absorption spectra of important undiluted ligands and all three copper (II) complexes in DMF at an ambient temperature. According to Figure 7a,b, prominent asymmetric absorption bands exhibit maxima at 280 nm and 294 nm, respectively, for π-π* electronic excitations in benzylimidazole and bipyridine. There is a slight shift of these absorptions to a lower energy when benzylimidazole and bipyridine are coordinated with copper (II), according to the red curves in Figure 7a,b. Similarly, for salicylic acid complexes with copper (II) in Figure 7c, there is a broad asymmetric absorption for the undiluted ligand at 286 nm with components of this absorption band at a lower energy in the copper (II) complex. Phenomenologically, there exists minor red-shifts for all three absorption bands in the presence of copper (II), according to the red curves in Figure 7a–c.

Figure 7d–f illustrates the two-dimensional excitation–emission correlations. Excitation on the vertical axis corresponds to absorption spectra in Figure 7a–c for each copper (II) complex. Hence, vertical slices at strong emission wavelengths in Figure 7d–f yield the absorption spectra in Figure 7a–c. Emission wavelengths on the horizontal axis are given as a function of the excitation wavelength. Fluorescent intensity contours increase from dark blue (i.e., zero) to red (i.e., maximum) [14]. 

The fluorescent emission of important undiluted ligands and all three copper (II) complexes in DMF at an ambient temperature is illustrated in Figure 8a–e. In each case, excitation induces a π-π* transition in the ligand, with subsequent ligand-to-metal energy transfer in the copper (II) complexes. A higher intensity emission occurs at longer wavelengths in all copper (II) complexes (i.e., red curves) relative to the emission in the corresponding undiluted ligands (i.e., black curves), according to Figure 8a–c. Results corresponding to the excitation and maximum emission intensity for three copper (II) complexes are summarized as follows:Benzylimidazole; excitation = 280 nm, ligand emission = 400 nm, copper (II) complex emission = 468 nmBipyridine; excitation = 294 nm, ligand emission = 425 nm, copper (II) complex emission = 485 nmSalicylic acid; excitation = 286 nm, ligand emission = 390 nm, copper (II) complex emission = 473 nm

Slit-width-corrected luminescent data in Figure 8d reveal that copper (II) complexes with salicylic acid exhibit at least a 4-fold increase in emission relative to the corresponding complexes with bipyridine or benzylimidazole, suggesting that copper (II) complexes with salicylic acid represent excellent candidates for photoactive materials that emit blue-green light. Figure 8f and Table 3 summarizes transient fluorescent decays, lifetimes, and quantum yields for all three copper (II) complexes in the solid state at an ambient temperature, in response to excitation at 306 nm.

Copper (II) complexes with salicylic acid exhibit the highest quantum yield, considering all three copper (II) complexes in this investigation (i.e., see Table 3) and all five copper (II) complexes studied previously (i.e., see Table 1). It is postulated that the sigma-donor and pi-donor characteristics of the carboxylic acid group in salicylic acid are responsible for efficient photo-emissions [38]. 

As shown in Figure 9, the absorption of the desired solid powder and the solid luminescence are given. In Figure 9a, the absorption range is not very different and is very similar. In Figure 9b,c, the solid luminescence and color pictures of the Complex A, B, and C powders are given. The wavelength of the complex emission and the color of the luminescence can be visually observed. In addition, by measuring the quantum yield of solid powder, we can find that the yield is C (30.75%) > B (13.35%) > A (6.8%). The powder precipitates in the liquid, and the concentration of the solid powder is increased, so the quantum yield is increased.

To further study the effect of Cu^2+^ content on the luminescence intensity of complexes, we changed the concentration of Cu^2+^ ions in each complex, and the results are shown in Figure 10a–c. It is noticeable that as the concentration increases, the luminescence intensity increases firstly, and then decreases. The complexes showed the strongest luminescence intensity when the concentration of Cu^2+^ was 0.02 mol/L. Excessive Cu^2+^ content causes the distance between centers to become a less than critical distance, which will produce a cascading energy transfer until finally entering an annihilation center to cause an energy transfer, which results in the annihilation of luminescence.

We also investigated the temperature influence on the emission of Cu^2+^ complexes from room temperature (295 K) down to 110 K. The temperature range is over a 185 K difference. As can be seen in Figure 10d–f, with the increase in temperature, the luminescent intensities exhibit a slight decline in intensity. On the other hand, there is a small, but apparent, spectral shift of the emission data upon decreasing the temperature. This phenomenon is generally expected for low-temperature experiments, resulting from thermal effects. At a low temperature, the Cu^2+^ atomic and ligand molecular thermal vibrations decreased and the ion-ligand aggregations became more condensed. All these occurrences cause less excitation energy loss and, thus, behave as a gradual blue-shift and a sharpening of the emission features. Based on the Figure 10g–i, we can try to simulate the data into a common equation, as follows:(1)$$I\left(T\right)=I_{0}+\frac{A}{T}$$
where T is the absolute temperature (K), I(T) is temperature-dependent luminescence intensity, and A is the temperature coefficient of luminescence intensity according to Equation (1), with dimensions of absolute temperature. Corresponding to Complex A, B, and C, this equation fits into the linear relationship in Figure 10g–i. The parameter values of I_0_ and A from the figures are listed in Table 4. R^2^ is the correlation coefficient. Based on Equation (1), the luminescence intensity has a linear relationship with the reciprocal of the temperature.

### 3.3. TG and Electrochemistry

The thermogravimetric analysis of complexes was carried out in the range of 18 °C to 900 °C in the presence of a nitrogen atmosphere with a heating rate of 10 °C min. As shown in Figure 11a, the initial mass loss of Complex A appeared at around 100 °C, which is mainly attributed to the evaporation of water molecules adsorbed at the surface of the complex. The major weight loss appeared at 230~500 °C range, which was caused by the pyrolysis of ligands sheets. For the coordination polymers B and C, there was no strictly clean weight loss step that occurred below 320 °C. The framework collapse and a sharp decomposition took place (a weight loss of 55%) from 320 to 600 °C, which can be assigned to the loss of ligands and phen [39,40,41]. The above-mentioned weight loss occurs later than in Complex A and it clearly proves that the thermal stability of complexes is in the order of C > B > A.

The electrochemical properties of the copper complexes were studied by cyclic voltammetry (CV). The voltammogram of complexes is shown in Figure 11b. Three complexes were redox active and displayed a quasi-reversible cyclic voltametric response. The voltammogram of the Cu^2+^ complexes indicated only one single reduction peak (Cu^2+^/Cu^+^) for each complex during the cathodic potential scan, which is located at −0.216 V, −0.212 V, and −0.048 V, respectively. During the return anodic potential scan, just after the reduction peak, an anodic peak is observed at 0.136, 0.097, and 0.278 V, respectively. The separation between the cathodic and anodic peak potentials ΔEp (ΔE = Epa − Epc) of 180 mV indicate a quasi-reversible redox process assignable to the Cu(II)/Cu(I) couple, and E_1/2_ [(Epa + Epc)/2] is equal to −0.095, 0.0575, and 0.115 V, respectively. The redox potential [A (−0.095) < B (0.0575) < C (0.115)] is attributed to the extension of the corresponding π framework around the Cu^2+^ center.

## 4. Conclusions

In conclusion, Cu^2+^ complexes were synthesized and characterized by FTIR, XPS, ^1^H NMR, UV-absorption, and luminescence emission. Luminescence studies of all three copper (II) complexes revealed blue-green emissions in each case. The highest quantum yield in DMF is achieved for copper (II) complexes with salicylic acid. Three ligands and their copper (II) complexes exhibited ambient-temperature luminescence, suggesting that these materials could be useful for luminescent devices. An increase in the concentration of copper (II) complexes in DMF reveals that maximum fluorescent emission occurs at 0.02 mole/Liter, prior to the onset of luminescence annihilation at higher copper (II) concentrations. There was a small but prominent spectral shift in the emission data upon decreasing the temperature that is generally expected for low-temperature experiments, resulting from thermal effects, including temperature-dependent solvent reorganization effects, causing a gradual blue-shift and a sharpening of the emission features. According to the literature, the UV–Vis absorption range of β-carotene is in the range of 400–550 nm, which is just within the range of the three complexes. Therefore, the complex solution can be made into a film, so that the luminescence is absorbed by β-carotene and, theoretically, the content of β-carotene can be increased, resulting in an increase in the yield of the carrot. After three months of observation on the synthesized copper complexes, the copper complexes have good chemical stability, the structure and the appearance remain unchanged, and the emission is stable. The complexes will have good application prospects in the fields of agriculture and marine science.

## Figures and Tables

**Figure 1 materials-15-01719-f001:**
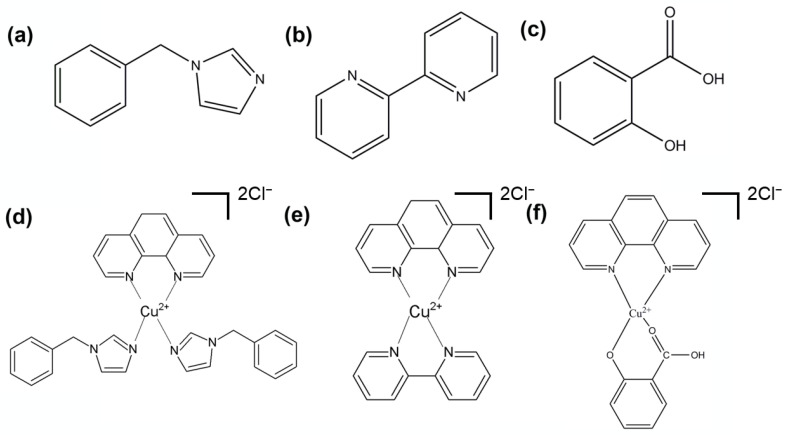
Structural chemical formulae of the three ligands (**a**–**c**) and its complexes (**d**–**f**).

**Figure 2 materials-15-01719-f002:**
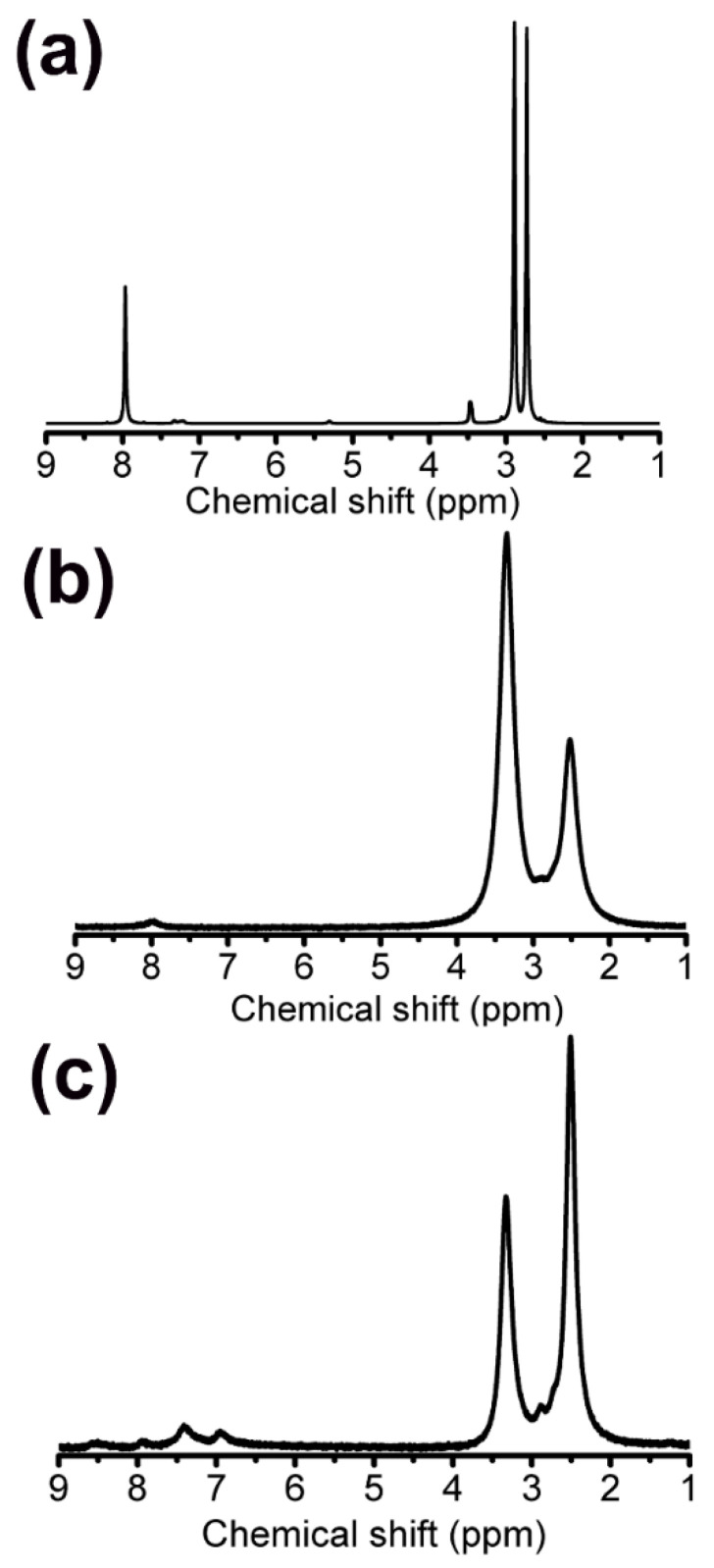
1H NMR of copper (II) complexes with (**a**) benzylimidazole, (**b**) bipyridine, and (**c**) salicylic acid. The concentration of all copper (II) complexes in NMR experiments is 10 mM.

**Figure 3 materials-15-01719-f003:**
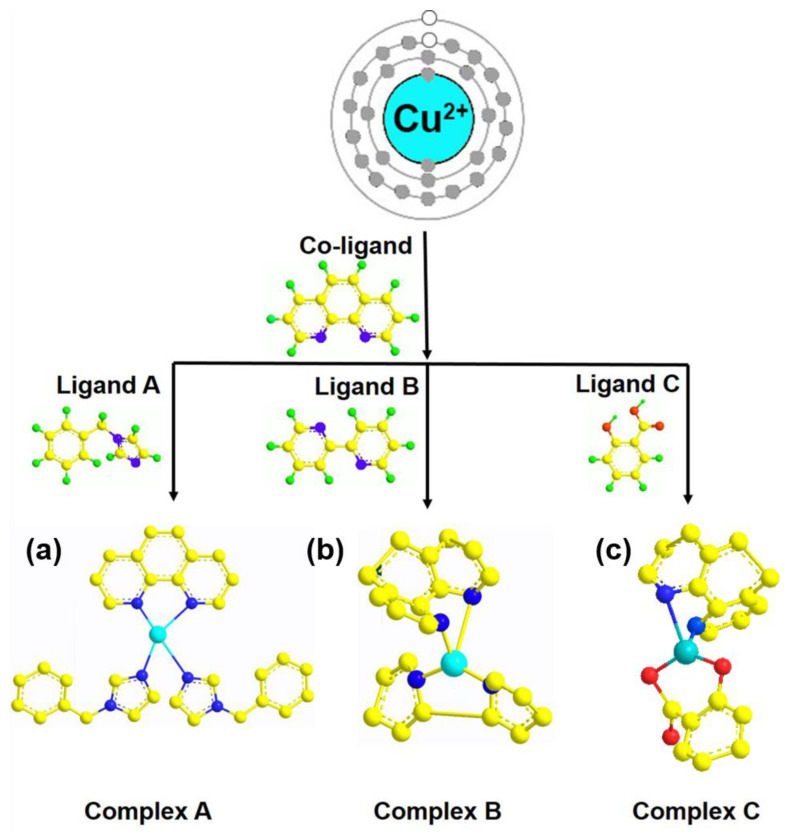
Schematic diagram of copper (II) complexes with (**a**) benzylimidazole, (**b**) bipyridine, and (**c**) salicylic acid.

**Figure 4 materials-15-01719-f004:**
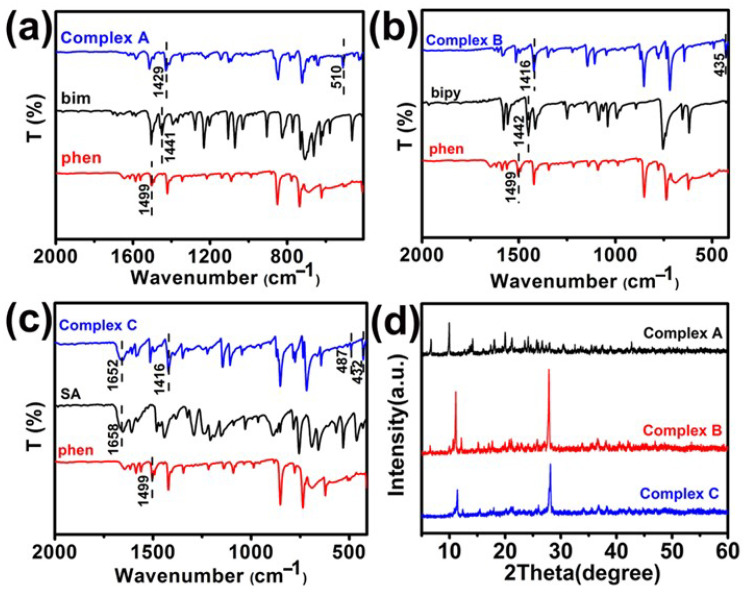
Illustration of FTIR spectra of copper (II) complexes with (**a**) benzylimidazole, (**b**) bipyridine, (**c**) salicylic acid, and (**d**) X-ray diffractograms of all three complexes using incident radiation at 1.54 Å.

**Figure 5 materials-15-01719-f005:**
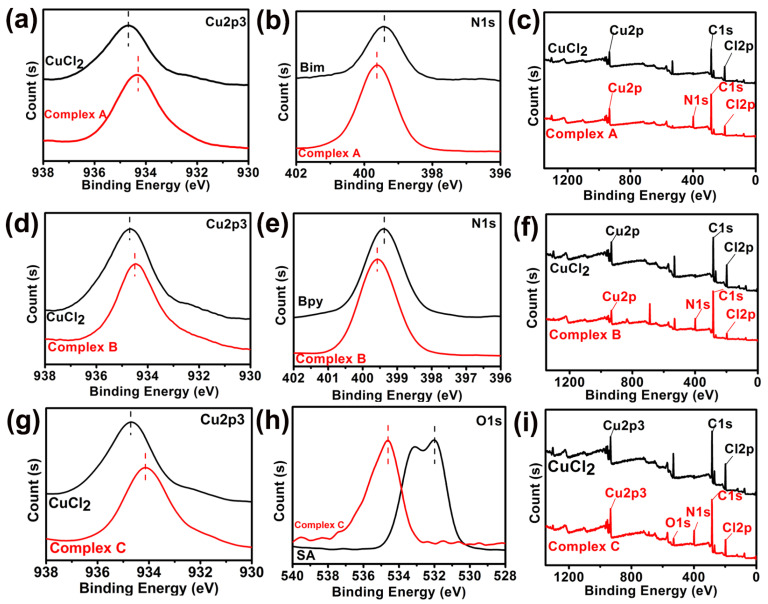
XPS data for Complex A: (**a**) copper 2p3/2, (**b**) nitrogen 1s, (**c**) XPS survey; Complex B: (**d**) copper 2p3/2, (**e**) nitrogen 1s, (**f**) XPS survey; Complex C: (**g**) copper 2p3/2, (**h**) oxygen 1s, (**i**) XPS survey.

**Figure 6 materials-15-01719-f006:**
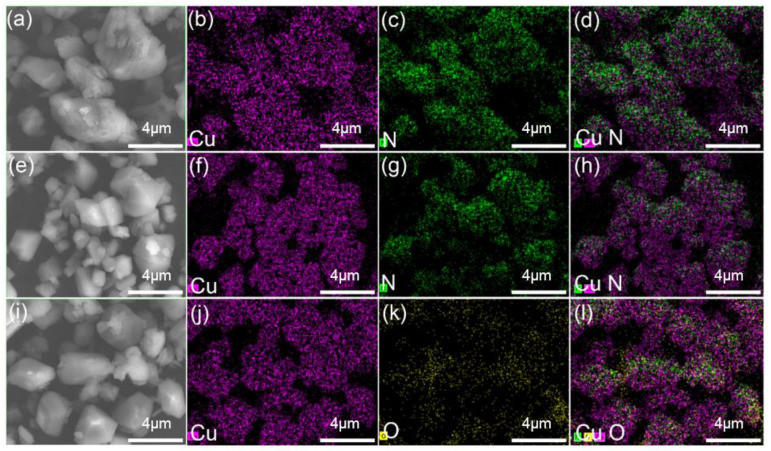
STEM dark-field (DF) images (**a**) SEM image of Complex A, (**b**–**d**) elemental mapping of Cu and N, (**e**) SEM image of Complex B, (**f**–**h**) elemental mapping of Cu and N, (**i**) SEM image of Complex C, (**j**–**l**) elemental mapping of Cu and O.

**Figure 7 materials-15-01719-f007:**
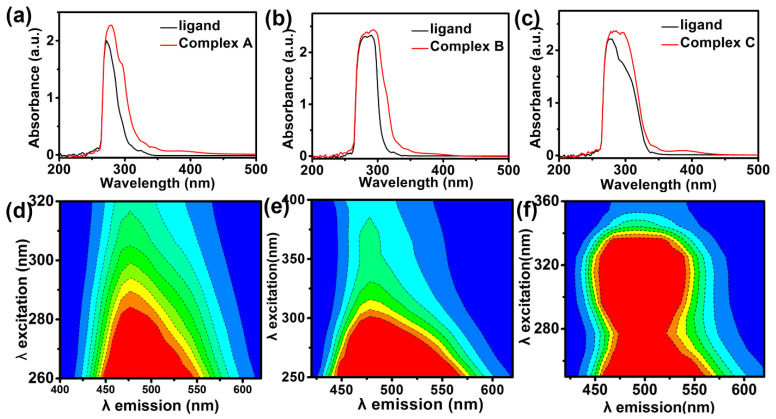
UV–visible absorptions of (**a**) benzylimidazole and its copper (II) Complex A, (**b**) bipyridine and its copper (II) Complex B, (**c**) salicylic acid and its copper (II) Complex C, and 2-dimensional excitation-emission spectra of (**d**) Complex A, (**e**) Complex B, and (**f**) Complex C.

**Figure 8 materials-15-01719-f008:**
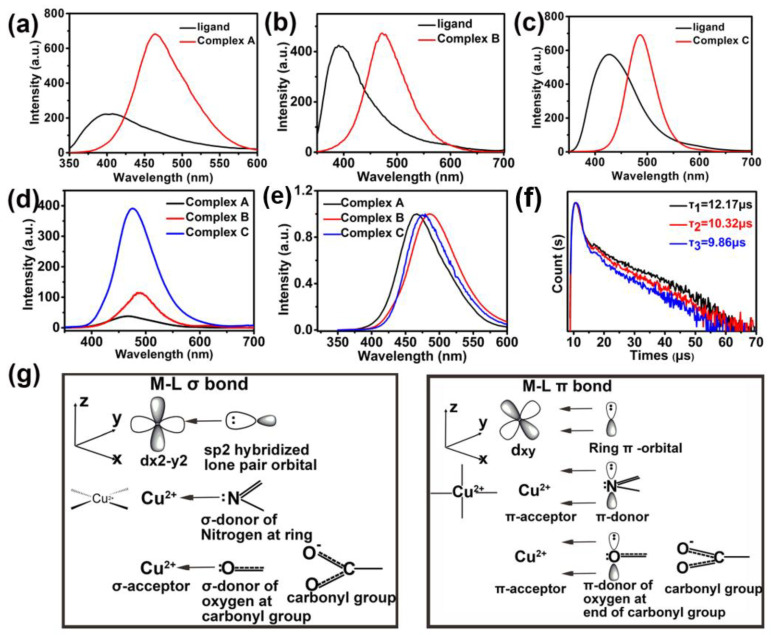
Fluorescence emission spectra for (**a**) benzylimidazole and its copper (II) Complex A (slit width = 20 × 20), (**b**) bipyridine and its copper (II) Complex B (slit width = 20 × 20), (**c**) salicylic acid and its copper (II) Complex C (slit width = 10 × 10), (**d**) comparison of luminescence intensity for all three copper (II) complexes (slit width = 10 × 10), (**e**) comparison of luminescence intensity for all three copper (II) complexes, each curve is normalized by its maximum intensity, (**f**) fluorescent decay curves for solid copper (II) complexes at the wavelength that corresponds to maximum emission intensity, and (**g**) molecular orbitals for copper (II) complexes subjected to Jahn–Teller distortions.

**Figure 9 materials-15-01719-f009:**
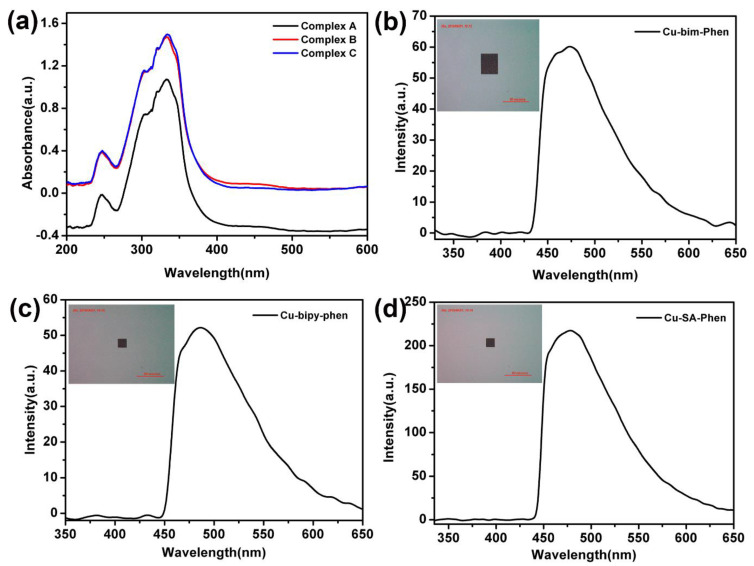
The absorption of all three copper (II) complexes (**a**) and solid luminescence of the copper (II) complexes with (**b**) benzylimidazole, (**c**) bipyridine, and (**d**) salicylic acid.

**Figure 10 materials-15-01719-f010:**
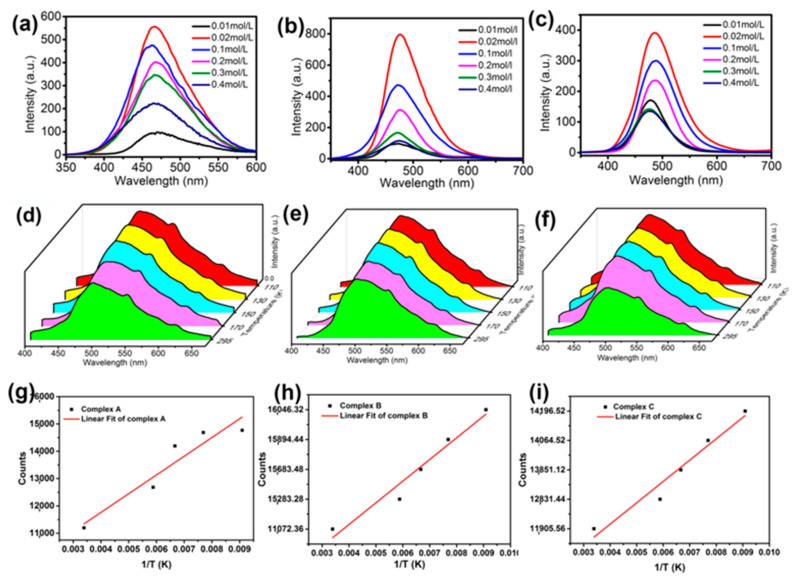
Concentration-dependent fluorescent emission in DMF at ambient temperature for copper (II) complexes with (**a**) benzylimidazole, (**b**) bipyridine, and (**c**) salicylic acid. Temperature-dependent fluorescent emission, between 110 K and 295 K for copper (II) complexes with (**d**) benzylimidazole, (**e**) bipyridine, and (**f**) salicylic acid. (**g**–**i**) Effect of temperature on maximum fluorescent intensity.

**Figure 11 materials-15-01719-f011:**
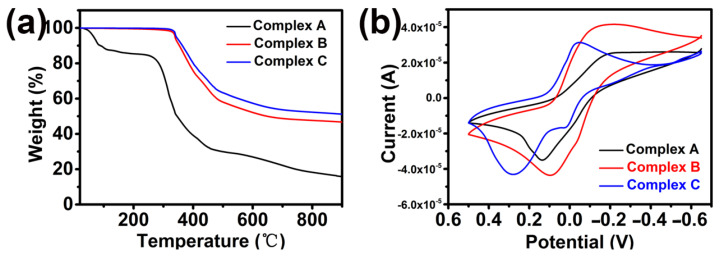
Illustrations of the (**a**) TG of the complexes and (**b**) cyclic voltammogram of complexes in KCl (solution 0.1 M) at 273 K using a glassy carbon working electrode, platinumwire auxiliary electrode, and an Ag/AgCl reference electrode; scan rate 100 mVs^−1^.

**Table 1 materials-15-01719-t001:** Published quantum yield lifetimes and peak emission wavelengths (nm) of Cu^2+^ complexes.

No.	Formula	Quantum Yield (%)	Lifetime (µs)	Peak Emission Wavelength (nm)	Ref.
1	[Cu_2_(opdc)_2_(phen)_2_] _n_	2.1	7.2 × 10^−4^	450	[25]
2	[Cu(bipy)(acac)(NO_3_)]	3.09	9.05	465	[29]
3	[Cu(bipy)(DMF)(NO_3_)_2_]	2.17	5.22	465	[29]
4	[Cu(bipy)(Ala)(NO_3_)(H_2_O)]	3.14	11.2	468	[29]
5	Cu(CDHPBC)	4.0	--	522	[31]

**Table 2 materials-15-01719-t002:** The elemental analysis of the copper complexes.

	Complex A		Complex B		Complex C	
	Mass Norm. (%)	Atom (%)	Mass Norm. (%)	Atom (%)	Mass Norm. (%)	Atom (%)
Copper	9.86	2.26	15.72	4.09	16.73	4.49
Nitrogen	12.75	13.28	12.06	14.24	10.84	13.21
Carbon	65.56	79.59	52.71	72.57	50.07	71.17
Chlorine	11.83	4.86	19.51	9.1	21.76	10.48
Oxygen	----	-----	----	----	0.6	0.64

**Table 3 materials-15-01719-t003:** Quantum yields and phosphorescence lifetimes of solid Cu^2+^ complexes.

Copper (II) Complex	Copper (II) Complex Structure	Quantum Yield (%)	Phosphorescence Lifetime (µs)	Emission Wavelength (nm)
A	Cu(bim)2(phen)	1.5	12.2	468
B	Cu(bipy)(phen)	4.9	10.3	485
C	Cu(SA)(phen)	11.9	9.9	473

**Table 4 materials-15-01719-t004:** Parameters for temperature-dependent luminescence, according to Equation (1).

Complex	I_0_	A (Kelvin)	R^2^
A = copper (II) with benzylimidazole	5.38	−638	0.87
B = copper (II) with bipyridine	7.7	−720	0.90
C = copper (II) with salicylic acid	7.74981	−725.82	0.945

## Data Availability

All data, models, and code generated or used during the study appear in submitted article.

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
