# Peer review of "An Efficient Cyan Emission from Copper (II) Complexes with Mixed Organic Conjugate Ligands"

_materials, 2022, doi:10.3390/ma15051719_

Round 1

Reviewer 1 Report

Dear Editor,

The manuscript with the ID materials-1587531 deals with the synthesis of Copper(II) complexes containing mixed ligand and their possible use as photoactive materials.

The work would seem interesting and suitable for publication with minor revision, but several aspects must be clarified.

  • ) what is the chemical stability over time of the solid state complexes? and how does the emission vary over time?
  • ) the authors mention the use of these complexes in the application field (agriculture) but it is not clear how. Could they describe in more detail this aspect?

Other minor remarks:

  • Line 55: missing a point
  • Line 64: to correct the caption in the table
  • Line 124 and 134: Were two different instruments used for FTIR analysis?
  • Line 187: to correct the format of the caption
  • Line 239: the reference to the figure seems wrong
  • Figure 8: reference to letter 8e and 8f is missing
  • Line 280: the content of the figure 9 should be explained in more detail in the text
  • Figure 10: the reference to letter 10d is missing
  • Line 300 and 306: superscript for +2
  • Table 10: subscript is missing

Author Response

Dear reviewers,

Thank you very much for your kindly comments on our manuscript. There is no doubt that these comments are valuable and very helpful for revising and improving our manuscript. In what follows, we would like to answer the questions you mentioned and give detailed account of the changes made to the original manuscript.

Question1. what is the chemical stability over time of the solid state complexes? and how does the emission vary over time?

Answer: Thank you very much for this kind of professional question.

Solid state complexes: for 3 months, within our observation time, it is stable with structure and the appearance keeps the same. We also believe that they are stable for long time.

The emission varies over time? The emission is also stable for 3 months within our observation time.

Question2. the authors mention the use of these complexes in the application field (agriculture) but it is not clear how. Could they describe in more detail this aspect?

Answer: Thank you very much for your pointing this out. We have the explanation from line 81-85, as below: “From an application viewpoint, cyan emission spectra overlap carotene absorption1.  ß-carotene is a key compound in grain plants, fruit trees, and other key food resource plants, which absorb solar energy at wavelengths between 400nm to 550nm.  Hence, the significance for agricultural harvest improvement is very important.”

We know the explanation is based on the basic knowledge from reference. The emission spectra of the three copper(II) complexes we obtained are between 430-550nm,which we can obtain from Fig9 of the paper. Therefore, the complex solution can be made into a film, adding the film to farm implements such as greenhouse, so that the luminescence can be absorbed by ß-carotene and theoretically, and the yield of the plants can be increased.

Figure from the reference 1

Fig9 from the paper

Question3. Other minor remarks:

  • Line 55: missing a point
  • Line 64: to correct the caption in the table
  • Line 124 and 134: Were two different instruments used for FTIR analysis?
  • Line 187: to correct the format of the caption
  • Line 239: the reference to the figure seems wrong
  • Figure 8: reference to letter 8e and 8f is missing
  • Line 280: the content of the figure 9 should be explained in more detail in the text
  • Figure 10: the reference to letter 10d is missing
  • Line 300 and 306: superscript for +2
  • Table 4: subscript is missing

Answer: Thank you for your significant reminding. According to your suggestion, we corrected the above grammatical errors and made effort to correct the spelling and grammar errors and polish the whole manuscript. We would like to confirm that the suitably revised manuscript is understandable to readers.

Thank you again for your positive and constructive comments and suggestions on our manuscript.

We hope you will find our revised manuscript acceptable for publication.

reference

  1. Alwis, D. D. D. H.; Chandrika, U. G.; Jayaweera, P. M., Spectroscopic studies of neutral and chemically oxidized species of beta-carotene, lycopene and norbixin in CH2Cl2: Fluorescence from intermediate compounds. Journal of Luminescence 2015, 158, 60-64.http://dx.doi.org/10.1016/j.jlumin.2014.08.036

Reviewer 2 Report

This well-performed work is devoted to the synthesis and detailed investigation of emissive   Cu(II) cationic complexes. It is generally accepted that the Cu2+ ion is a strong emission quencher, but the results of this work reveal that Cu(II) based complexes can be luminescent ones. In my opinion, the findings presented in reviewed work are interesting for researchers dealing with luminescent materials. Therefore, I recommend acceptance this paper after revision to address the following concerns. 
1. In the Introduction, the phrase “Although some examples of complexes with high-spin ground states and ordered structures have been achieved, …” should be better supplied by appropriate references, e.g. on Mn(II) complexes (DOI: 10.1039/D1NJ02053F, 10.1039/D1NJ02053F, 10.1039/C9DT03283E, 10.1016/j.cej.2021.129886, 10.1016/j.inoche.2019.107473). 
2. What are the structures drawn in Figure 3? - Please, add some comments in legends. In Fig. 1, the counter-ions should be added to the cationic species. 
3. If possible, X-ray structures of the compounds obtained should be provided. 
4. The lifetimes listed in Table 3 are too long to be considered fluorescent ones. The lifetimes of such order are more typical for phosphorescence rather than fluorescence. 
5. In the Experimental part, the preparative yields of the products (in mg) should be given along with the formulation of the complexes.

Author Response

Dear reviewers,

Thank you very much for your kindly comments on our manuscript. There is no doubt that these comments are valuable and very helpful for revising and improving our manuscript. In what follows, we would like to answer the questions you mentioned and give detailed account of the changes made to the original manuscript.

Question1. In the Introduction, the phrase “Although some examples of complexes with high-spin ground states and ordered structures have been achieved, …” should be better supplied by appropriate references, e.g. on Mn(II) complexes (DOI: 10.1039/D1NJ02053F, 10.1039/C9DT03283E,10.1016/j.inoche.2019.107473). 

Answer: Thank you for your introduction to these wonderful research work. According to your suggestion, we properly cite these articles as reference2-5 in line 33 of the manuscript.

Question2. What are the structures drawn in Figure 3? - Please, add some comments in legends. In Fig. 1, the counter-ions should be added to the cationic species

Answer: Thanks for your professional suggestion. (1) I have added comments in legends between line 149 to line 152 – “Figure 3 is a schematic diagram of the coordination of three different copper(II) complexes. All three copper complexes have two kinds of ligands, the first is the co-ligand: phen, and then the species of the second ligand is changed, which is (a) benzylimidazole, (b) bipyridine, (c) salicylic acid, thereby forming three new copper(II) complexes.”

(2) I have added the cationic species to the Figure 1 and made corrections in the manuscript.

Question3. If possible, X-ray structures of the compounds obtained should be provided. 

Answer: Thanks for your valuable advice. We understand that XRD may better reveal the structure of the copper complexes. You are right. Since, we previously focused on the luminous property of copper complexes, and currently university in vocation, the campus is at closing status. Surely, we will do this in the coming research. At the revision period, it is not easy to do.

Question4. The lifetimes listed in Table 3 are too long to be considered fluorescent ones. The lifetimes of such order are more typical for phosphorescence rather than fluorescence. 

Answer: Thanks for your valuable counsel. According to your suggestion, we made corrections in the manuscript.

Question5. In the Experimental part, the preparative yields of the products (in mg) should be given along with the formulation of the complexes.

Answer: Thank you very much for your nice suggestion. Through your important advice, we have given the preparation yields (in mg) of three different copper complex products on lines 102, 111 and 119 of the experimental section.

Thank you again for your positive and constructive comments and suggestions on our manuscript.

We hope you will find our revised manuscript acceptable for publication.

Reviewer 3 Report

The manuscript materials-1587531 "PEfficient Cyan Emission from Copper (II) Complexes with Mixed Organic Conjugate Ligands" by Tang and co-workers describes the synthesis of three copper(II) complexes and the study of their properties. The synthesis of copper(II) complexes confirmed by FTIR, 1H NMR, XPS, TG/DTA, SEM, EDX, UV-Vis and fluorescence spectroscopy.

Questions and comments:

1) I recommend that the authors strengthen the Introduction part on the design of copper (II) complexes especially with high thermal stability. New articles on the design of these copper (II) complexes, as well as their applications, should be added. For example, Molecules 2021, 26(8), 2334; Mol. Struct. 2021, 1224, 129069; J. Energ. Mater. 2020, 39(1), 23-32.

2) Why are the aromatic ligands nonplanar in Figure 1? What method was used to confirm such structures of the complexes? The coordination in Figures 1f and 3 (complex C) does not match.

3) What concentration of the complexes was used? Was there any influence of the paramagnetic properties of the copper (II) cation on the NMR spectra? The NMR spectra show traces of solvents. Based on the proposed structures of the complexes, only the signals of aromatic protons should be observed in the 1H NMR spectra. Figure 2a shows the 1H NMR spectrum of DMF with some impurities. How can the authors explain the peaks around 2-3 ppm?

4) How can the authors explain the difference between the found and calculated С (%) in elemental analysis of complexes A and C?

5) Scale bars should be added to Figure 6.

6) How and for how long were the obtained complexes dried? How was the purity of the complexes confirmed?

7) Did the authors evaluate the binding constants of the obtained complexes?

8) The manuscript should be checked for errors (for example, line 349, uv-absorption; lines 300, 306, Cu2+ etc). Lines 167-170, did the authors confuse Figures 3 and 4? English also should be re-checked.

9) I recommend that the authors add a clearer and more detailed description of the possible application of the obtained results in conclusion.

Author Response

Dear reviewers,

Thank you very much for your kindly comments on our manuscript. There is no doubt that these comments are valuable and very helpful for revising and improving our manuscript. In what follows, we would like to answer the questions you mentioned and give detailed account of the changes made to the original manuscript.

Question1. I recommend that the authors strengthen the Introduction part on the design of copper (II) complexes especially with high thermal stability. New articles on the design of these copper (II) complexes, as well as their applications, should be added. For example, Molecules 202126(8), 2334; Mol. Struct. 2021, 1224, 129069J. Energ. Mater. 2020, 39(1), 23-32.

Answer: Thank you for your introduction to these wonderful research work. According to your suggestion, we properly cited these articles as reference16-18 in line 42 of the manuscript.

Question2. Why are the aromatic ligands nonplanar in Figure 1? What method was used to confirm such structures of the complexes? The coordination in Figures 1f and 3 (complex C) does not match.

Answer: Thanks for your valuable suggestions. (1).The nonplanar structure is just generated by Chem3D, (2) We used the method of EDS to confirm the ratio of elements , and further to determine the ratio of ligand numbers. (3) I have corrected the C complex in Figure 3.

Question3: What concentration of the complexes was used? Was there any influence of the paramagnetic properties of the copper (II) cation on the NMR spectra? The NMR spectra show traces of solvents. Based on the proposed structures of the complexes, only the signals of aromatic protons should be observed in the 1H NMR spectra. Figure 2a shows the 1H NMR spectrum of DMF with some impurities. How can the authors explain the peaks around 2-3 ppm?

Answer: Thanks for your professional question. (1) The concentration of the complexes was 10 mM. (2) We checked the relevant literature and concluded that the NMR method is a very common method for the determination of metal complexes. The paramagnetic properties of copper will slightly affect the NMR test results, so we will support our experimental conclusions through the data results of FTIR, EDS and other characterization methods. But we are also aware of the shortcomings of this work and will continue to study in future work. (3) We provided a solid sample when doing MNR testing, and DMF is used as a solvent for laboratory testing, so there is a corresponding peak at 2-3ppm in the test results.

Question4: How can the authors explain the difference between the found and calculated С (%) in elemental analysis of complexes A and C?

Answer: Thank you very much for pointing this out. After seriously considering your question, we re-calculated the experiment and revised the experimental data. Thank you very much for your valuable comments, and we will continue to study this problem in future work.

Question5. Scale bars should be added to Figure 6.

Answer: Thank you very much for the correction, I am sorry for my omission and have corrected it in the manuscript.

Question 6. How and for how long were the obtained complexes dried? How was the purity of the complexes confirmed?

Answer: Thank you very much for your professional question. (1) The copper complex obtained by drying under vacuum for 48 hours. (2) We are currently using the elemental analysis method to obtain the purity of the copper complex by calculating the ratio between the data measured by EDS and the theoretical value.

Question7. Did the authors evaluate the binding constants of the obtained complexes?

Answer: Thank you very much for pointing this out, but sorry Prof we did not calculate the binding constant. Because the binding constant is usually considered to under liquid conditions; and our complexes are usually used as luminescent materials at solid conditions, so they are not estimated.

Question8.The manuscript should be checked for errors (for example, line 349, uv-absorption; lines 300, 306, Cu2+ etc). Lines 167-170, did the authors confuse Figures 3 and 4? English also should be re-checked.

Answer: Thank you for your significant reminding. According to your suggestion, we corrected the above grammatical errors and made effort to correct the spelling and grammar errors and polish the whole manuscript. We would like to confirm that the suitably revised manuscript is understandable to readers.

Question9.I recommend that the authors add a clearer and more detailed description of the possible application of the obtained results in conclusion.

Answer: Thank you for your important reminder. Based on your suggestion, we have added more literature to illustrate the paper, further polishing the article.

Thank you again for your positive and constructive comments and suggestions on our manuscript.

We hope you will find our revised manuscript acceptable for publication.

Round 2

Reviewer 3 Report

I thank the authors for improving this manuscript. Please, add information about the concentration of copper(II) complexes in NMR experiments to the manuscript text.

However, I have one question. The authors state that "We are currently using the elemental analysis method to obtain the purity of the copper complex by calculating the ratio between the data measured by EDS and the theoretical value." The difference calc (exp) for C atoms in complexes B and C is almost 4%, which is unacceptable in the elemental analysis. I ask the authors to double-check it and/or give an explanation of such an effect in the manuscript text.

Author Response

Dear reviewer,

Thank you very much for taking time of your busy schedule to review our manuscript. Now we have carefully corrected and replied the manuscript for the revision. The revision instructions are as follows:

Question1.I thank the authors for improving this manuscript. Please, add information about the concentration of copper(II) complexes in NMR experiments to the manuscript text.

Answer: Thank you very much for pointing this out. We have added the concentration of copper complexes in NMR experiments to the manuscript text in line 144.

Question2.The difference calc (exp) for C atoms in complexes B and C is almost 4%, which is unacceptable in the elemental analysis. I ask the authors to double-check it and/or give an explanation of such an effect in the manuscript text.

Answer: After seriously considering your question, we re-calculated the experiment and revised the experimental data. The purity of the complex is indeed a constant concern for us. In this article, we mainly focus on the synthesis of new copper complexes and support them by methods of NMR, FTIR, XPS, etc. We temporarily use EDS to roughly estimate the purity of the complexes, however, due to factors such as the scanning area of EDS, it is not the optimal test method in quantitative calculation, and there are deviations. Therefore, we are also constantly optimizing the experiments to improve the purity, and looking for newer testing methods to further measure the purity, and will continue to do the same in future research work. We are very sorry that due to the epidemic, we cannot return to school for further data supplementation, but we will improve this work in future research.

Thank you again for your positive and constructive comments and suggestions on our manuscript.

We hope you will find our revised manuscript acceptable for publication.